# The Effects of Decreasing Foot Strike Angle on Lower Extremity Shock Attenuation Measured with Wearable Sensors

**DOI:** 10.3390/s25092656

**Published:** 2025-04-23

**Authors:** Lucas Sarantos, David J. Zeppetelli, Cole A. Dempsey, Takashi Nagai, Caleb D. Johnson

**Affiliations:** 1Military Performance Division, U.S. Army Research Institute of Environmental Medicine, 10 General Greene Ave., Natick, MA 01760, USA; 2Oak Ridge Institute for Science and Education, 100 Orau Way, Oak Ridge, TN 37830, USA

**Keywords:** biomechanics, shock attenuation, inertial measurement units, gait

## Abstract

Shock attenuation may be a clinically feasible method to assess changes in lower extremity joint loading induced by gait modifications, such as decreasing foot strike angle (forefoot striking). The purpose of this study was to identify changes in lower extremity shock attenuation between habitual and forefoot strike gait conditions. Eighteen participants ran on a treadmill with their habitual gait and an instructed forefoot strike gait. Shock attenuation was measured with inertial measurement units as the ratio of proximal to distal peak resultant/vertical accelerations, with three sensor combinations: ankle to below/above knee (BK/A; AK/A) and AK/BK. Three participants were excluded who were habitual forefoot strikers or failed to decrease their foot strike angle by at least 5° in the forefoot strike condition. The results showed significantly greater resultant shock attenuation in the forefoot strike compared to the habitual condition for BK/A (mean Δ = 0.13, *p* = 0.004) and AK/A (mean Δ = 0.23, *p* = 0.007). No significant differences were found for AK/BK or vertical shock attenuation. These results suggest that shock attenuation may not reflect joint-specific loading changes that have been shown for forefoot striking (i.e., increased ankle/shank and decreased knee moments). However, it may capture changes in overall lower extremity loading (i.e., decreased vertical ground reaction forces).

## 1. Introduction

Overuse musculoskeletal injuries (MSKIs) related to running affect both civilian and military populations. In civilians, rates of injury in recreational runners have been reported to be as high as ≈90% for one-year incidence and more consistently between 34 to 50% [1,2]. In turn, the incidence of MSKI in the military has been reported to be 42/63% in men/women during just initial-entry training (10-week period) [3,4]. Further, 75% of total MSKIs during initial-entry training are reported to be overuse in nature [5], and running is consistently reported as one of the most common activities leading to injury [6,7]. One contributing factor to many running-related MSKIs is increased lower extremity loading. This includes increased overall ground reaction forces (GRFs) or rates of GRF application [8,9], and joint-specific forces, or moments [10,11]. One method that has been promoted for reducing lower extremity loading during running, and thereby MSKI risk, is having an individual decrease their foot strike angle, also known as a forefoot striking.

It has been consistently demonstrated that vertical GRFs and loading rates can be successfully decreased by changing from a natural rearfoot (landing more on the heel) to forefoot strike pattern, or decreasing the foot strike angle to land more on the ball of the foot (Figure 1) [12,13]. Additionally, several studies have shown that a habitual or imposed forefoot strike pattern tends to decrease knee joint moments or loading and increase moments/loading about the ankle and shank [13,14,15,16]. Previous studies have shown that the overwhelming majority of runners (95%) [17] and military personnel (87%) [18] run with a habitual rearfoot foot strike angle. Given this, decreasing foot strike angle during running may be a promising treatment method for individuals with running-related MSKIs at the knee, which is consistently the most common site for overuse injuries [1,2,5]. However, capturing changes in lower extremity loading during running requires force plates and three-dimensional motion capture. This makes it unfeasible for a typical clinical environment, where access to this technology, and the technical expertise needed to operate it, is usually not available. Alternatively, shock attenuation may offer a more “clinically feasible” method of capturing joint-specific changes in loading as a result of forefoot striking and other gait modifications.

Shock attenuation describes the way in which the joints and musculoskeletal structures attenuate the forces imparted to the body during running, starting from the foot and traveling up through the kinetic chain to the head [19]. Shock attenuation has been calculated as the change in resonant frequency of body segment accelerations [19] as well as the reduction in peak accelerations [20]. Further, with advancements in wearable sensors containing accelerometers, shock attenuation can be captured easily in a clinical environment by attaching small, wireless sensors onto the segments of interest [20,21]. Previous work has focused on full-body shock attenuation, from the approximate point of origin of GRFs (just above the ankle) to the head [19,21,22]. One of these studies reported significant differences in full-body shock attenuation between habitual rearfoot and forefoot strike runners [22]. However, no previous studies have examined shock attenuation within the lower extremities (i.e., local shock attenuation) or the effects of altering foot strike angle to impose a forefoot strike pattern on shock attenuation. Given the previous research demonstrating that forefoot striking may decrease knee moments but increase ankle/shank moments [13,14,15,16], changes in local shock attenuation with this gait modification, below and above the knee, are of particular importance to investigate. Logically, one might expect shock attenuation below the knee to increase with forefoot striking but decrease above the knee, mirroring the changes in moments seen in previous work. However, these specific changes in local shock attenuation with forefoot striking have yet to be investigated.

The purpose of the current study was to examine the effects of decreasing foot strike angle (i.e., switching to a forefoot strike pattern) on shock attenuation from the ankle to below and above the knee. Given previous work on the effects of decreasing foot strike angle on joint-specific loading, we hypothesized that decreasing foot strike angle would result in greater shock attenuation below the knee, but lower shock attenuation above the knee. If the results are as expected, this will provide initial evidence that shock attenuation may be able to capture similar effects as those demonstrated for joint-specific loading measures, which require expensive/complex equipment and procedures. Given the better accessibility of shock attenuation as a measure, this would provide practitioners with a more viable method of tracking the effects of gait modifications, like decreasing foot strike angle.

## 2. Materials and Methods

### 2.1. Study Design and Participants

This study utilized an observational, cross-sectional design, with within-subject comparisons. The current study is a secondary analysis of data from a larger study on the effects of several gait modification techniques on running and load carriage mechanics. All procedures were reviewed and approved by the U.S. Army Medical Research and Development Command Institutional Review Board and written informed consent was obtained. A total of 18 healthy participants were recruited and met the following criteria: ages 18–40 years, physically active [23], without active or previous injuries that would prevent one from exercising, not currently pregnant, and without any health conditions or taking medications that would alter balance or affect the ability to exercise.

### 2.2. Procedures

All testing was completed in one session. Participants were first outfitted with inertial measurement units (IMUs) (Ultium Motion, Noraxon USA, Scottsdale, AZ, USA), placed over the distal–medial shank (Figure 2; just above the malleoli, over the tibia), proximal–medial shank (just below the patella, over the tibia), and distal–medial thigh (just above the knee, on the flattest/most superficial part of the femur, in line with the proximal shank sensor). These IMUs (weight = 19 g, dimensions = 1.22 cm (H), 4.45 cm (L), 3.3 cm (W)) include an accelerometer (range = ±16 g) and gyroscope (range = ±7000 deg/s), both with sampling rates of 200 Hz. The sensors were secured with adhesive spray and double-sided tape, followed by pre-wrap and Coban tape over top of the sensors. Participants were also fitted with 56 retroreflective markers, using a previously described marker set [24]. For the current study, the relevant markers were those used to track the foot segment (and calculate foot strike angle) including markers placed on the head of the 1st/5th metatarsals, the head of the 2nd proximal phalanx, the lateral heel, and the proximal/distal posterior heel.

A standing calibration trial was collected for later use in building a kinematic model. Two motion calibration trials were also collected for use in standardizing the orientation of the IMU accelerometer axes, where the participant performed three repetitions of knee and hip flexion/extension. All testing was performed on an instrumented treadmill (Tandem Treadmill, AMTI, Watertown, MA, USA; sampling rate = 1200 Hz), and marker trajectory data were collected utilizing a 16-camera optical motion capture system (Qualisys, Gothenburg, Sweden; sampling rate = 200 Hz). Participants were first given a five-minute warmup at a self-selected comfortable running pace. To establish the pace, the treadmill was started at 2.24 m/s. The speed was then gradually increased (≈0.09 m/s increments) until a comfortable training pace for the participant was reached, defined as one that felt natural but that they could easily maintain for 20 min (mean pace = 2.38 ± 0.31 m/s). Following the warmup, two minutes of data were collected (habitual gait condition). Participants were then instructed to decrease their foot strike angle by running more on the front or ball of their foot and raising their heel more at foot strike. They were given one minute to practice the new pattern, with verbal feedback provided, followed by two minutes of data collection (forefoot strike condition). Ground reaction force and marker trajectory data were collected with Qualisys QTM (v.2024.3) and time-synchronized with IMU data using Noraxon’s MR4 software and myoSync hardware.

### 2.3. Data Reduction and Analysis

Data reduction was performed using Qualisys QTM, Visual 3D (v.2024.06.1, HAS-Motion, Kingston, ON, Canada) and custom MATLAB (v. R2024a, MathWorks, Natick, MA, USA) scripts. For IMU data, the accelerometer axes were first reoriented into a body-fixed frame (i.e., relative to the path of motion of the leg) using the two motion calibration trials and procedures first reported by Cain et al. [25] and described previously [26]. The vertical axis of the accelerometer was defined as being in line with gravity. Resultant accelerations were calculated as the root sum of squares of accelerations along all three accelerometer axes. Using GRF data, the stance phase for each stride was identified with a threshold of 40 N for foot strike/toe off events, similar to values in previous studies utilizing instrumented treadmills [27,28]. Peak vertical and resultant accelerations during stance were extracted (Figure 3) for each stride and IMU, and shock attenuation was calculated as the proximal over distal IMU ratio. Therefore, lower values indicated greater attenuation, and values over one indicated that peaks were higher at the proximal IMU versus distal (i.e., peaks were not attenuated). Shock attenuation was calculated for three combinations of sensors: below knee/ankle (BK/A), above knee/ankle (AK/A), and above knee/below knee (AK/BK).

Sagittal plane foot angles were calculated from marker trajectory data, as described previously [24], with zero degrees indicating a flat foot and negative/positive values indicating a plantar/dorsiflexed foot. Foot angles were low-pass filtered (4th-order, cutoff frequency = 10 Hz), and the angle at initial contact (see above) for each stride was extracted as the foot strike angle. For all variables, the average of 20 strides for the right leg was used for data analysis, which has been shown to produce reliable running biomechanical data [29,30].

IBM SPSS (v 28.0.1.1, Armonk, NY, USA) was used for data analysis. Foot strike angle was first inspected to exclude participants from analyses who (a) were habitual forefoot strikers, defined from previous work [31] as a foot strike angle of ≤−1.6°, or (b) did not decrease their average foot strike angle by at least 5° in the forefoot strike condition, compared to habitual. Two-way repeated measures analysis of variance (ANOVA; 3x2) was used to test the main and interactive effects of sensor location and gait condition on resultant and vertical peak shock attenuation. Mauchly’s test was used to check for sphericity, and Greenhouse–Geiser corrections were used where sphericity was violated. Pairwise t-tests were used to perform post hoc comparisons, reported with Bonferroni-adjusted *p*-values. An alpha level of 0.05 was used for all tests.

## 3. Results

Three participants were excluded from analyses (habitual forefoot striker = 2, failed to meet forefoot strike condition = 1), leaving fifteen to be included in the final analyses (12/3 male/female, age = 26.53 ± 6.07 yrs., height = 1.72 ± 0.068 m, weight = 76.93 ± 14.72 kg). The mean change in foot strike angle (forefoot habitual condition) was −16.1 ± 7.9°.

### 3.1. Resultant Acceleration Results

The results for resultant peak acceleration shock attenuation are displayed in Figure 4. Significant main effects of sensor location (F = 34.54, *p* < 0.001, ηp^2^ = 0.71) and gait condition (F = 8.92, *p* = 0.010, ηp^2^ = 0.39) on resultant shock attenuation were found. Pairwise tests showed significantly lower shock attenuation for AK/BK compared to BK/A and AK/A (mean differences = 0.36 and 0.31, *p* < 0.001). The habitual gait condition showed significantly lower shock attenuation compared to forefoot strike (mean difference = 0.16, *p* = 0.01). While the omnibus test for the interaction effect of sensor location x gait condition on resultant shock attenuation was not significant (F = 1.01, *p* = 0.338, ηp^2^ = 0.07), there were significant pairwise differences. The forefoot strike condition showed greater shock attenuation for BK/A (mean difference = 0.13, *p* = 0.007, d = 0.81) and AK/A (mean difference = 0.23, *p* = 0.004, d = 0.89) compared to habitual, but no difference for AK/BK (*p* = 0.275). Post hoc power for the omnibus test for the interaction effect was low (β = 0.16).

### 3.2. Vertical Acceleration Results

The results for vertical peak acceleration shock attenuation are displayed in Figure 5. A significant main effect of sensor location (F = 14.35, *p* < 0.001, ηp^2^ = 0.51) on vertical shock attenuation was found. Identical to resultant accelerations, pairwise tests showed lower shock attenuation for AK/BK compared to BK/A and AK/A (mean differences = 0.23 and 0.18, *p* = 0.006 and <0.001). The main effect of sensor location (F = 0.76, *p* = 0.399, ηp^2^ = 0.05) and the interaction effect (F = 0.09, *p* = 0.811, ηp^2^ = 0.01) were not significant, including in post hoc comparisons (*p* = 0.399–0.579).

## 4. Discussion

The purpose of the current study was to examine changes in lower extremity shock attenuation with an imposed decrease in foot strike angle compared to habitual gait. We expected that shock attenuation would be greater below the knee and reduced above the knee in the forefoot strike condition (i.e., decreasing foot strike angle).

Our primary finding was that decreasing foot strike angle resulted in increased shock attenuation for peak resultant accelerations from the ankle to both below and above the knee. Further, these differences were associated with large effect sizes [32]. This was partially in line with our hypotheses, which were based on previous work demonstrating reduced joint moments at the knee but increased moments below the knee associated with a forefoot strike pattern [13,14,15,16]. However, it is also true that overall lower extremity loading (i.e., GRFs) has been shown to be reduced with a forefoot strike gait modification [12,13]. Therefore, the increased shock attenuation below and above the knee may simply reflect the effect of decreasing one’s foot strike angle during running on overall reductions in lower extremity loading.

This is the first study to address the effects of decreasing foot strike angle or forefoot striking on shock attenuation at the lower extremity level. One study by Gruber et al. [22] found significantly greater total shock attenuation from the ankle to the head in runners with a habitual rearfoot strike vs. forefoot strike pattern. One explanation for these differences in findings, compared to the current study, is the differences between imposed and habitual forefoot striking, especially with the acute changes examined in the current study. It has been shown that runners acutely changing from a rearfoot to forefoot strike pattern are able to, largely, replicate the mechanics seen in habitual forefoot strikers [14]. However, several variables, including ankle and knee moments, remained significantly different in an imposed vs. habitual forefoot strike pattern [14]. Alternatively, it may be that rearfoot striking induces greater shock attenuation at the hip level and above, resulting in greater overall shock attenuation from the ankle to the head. Future work should focus on capturing shock attenuation through the full kinetic chain (ankle to head), as well as with a habitual pattern vs. imposed forefoot strike gait modification, which may elucidate the differences between the current and previous study findings.

We also found no differences in attenuation of vertical peak accelerations between gait conditions. This is surprising given that forefoot striking has been associated with lower vertical peak tibial accelerations (identical to site for “ankle” in current study) [33]. Further, peak vertical tibial accelerations have shown consistent, strong correlations with vertical GRF loading rates [29,34], which in turn have been shown to be significantly reduced with forefoot striking [12,13]. Despite this, our results show that attenuation of vertical peak accelerations at the lower extremities is not affected by decreasing foot strike angle. It is difficult to determine the reason for this. However, one explanation may be that the alignment of the vertical axis of the sensors at or just after foot strike differs between gait conditions due to differences in the angle of the segments they were attached to. This is a possibility even with the reorientation procedure used [25], as this would only align the axes at rest. It may be possible to align the axes of sensors continuously throughout a given movement pattern like a running stride. However, this would involve embedding the sensors in the motion capture global coordinate system, which we were not able to do, or more complex sensor fusion techniques. While these techniques may show different results in vertical shock attenuation from the current study, they would also limit the generalizability to clinical or other real-world applications. Therefore, our results suggest that resultant acceleration shock attenuation, which is agnostic to the relative alignment of sensor axes, should be the primary variable or outcome used when assessing the effects of gait modifications, like decreasing foot strike angle.

Our last important finding was that shock attenuation, across both modifications and for vertical/resultant accelerations, was greater for BK/A and AK/A compared to AK/BK. More importantly, the average shock attenuation ratios for AK/BK were over one, indicating that peaks were actually greater for AK compared to BK. It is difficult to hypothesize the reason for this effect, as logically one would expect peak accelerations to decrease as the measurement site moves proximally. There was significant variability in the mean effect, with many participants exhibiting attenuation ratios below one. Further, it could be that the thigh is less constrained, compared to the shank/ankle, throughout the stance phase of running, allowing for greater peak accelerations. Finally, it may be that our method of calculating shock attenuation was not sensitive to changes between measurement sites with such close proximity directly above and below the joint. We chose to calculate shock attenuation as the reduction in peak accelerations given its simplicity and, therefore, alignment with the goal of establishing a measure with greater clinical feasibility. However, previous studies have also calculated the change in resonant frequency of segmental accelerations [19,21], which may offer a more sensitive measure for shock attenuation between sites with close proximity.

There are some limitations to the current study. As eluded to earlier, we only captured acute or immediate changes in shock attenuation resulting from instructions to decrease foot strike angle. While previous work has shown that significant changes in gait mechanics can be achieved with acute changes in foot strike angle, our results cannot speak to the effects of more prolonged habituation to this gait modification or others. Second, previous work has shown significant effects of increasing running speed on shock attenuation [35]. Given that this was a secondary analysis of data, we were not able to examine whether the effects of decreasing foot strike angle on shock attenuation were consistent across higher running speeds. Briefly, it may be that the effects of increasing running speed outweigh or overshadow those of decreasing foot strike angle, and this should be investigated in future work.

In summary, this was the first study to examine the effects of a gait modification aimed at reducing lower extremity loading on local shock attenuation, specifically through the lower extremities. We found that decreasing foot strike angle led to greater shock attenuation, both below and above the knee. While this supports the use of this measure to track changes in overall lower extremity loading, it does not appear that changes in shock attenuation mirror those reported for joint-specific loading. Future work should focus on confirming these results relative to longer-term changes in gait, with more time for practice and habituation, as well as across increased running speeds.

## Figures and Tables

**Figure 1 sensors-25-02656-f001:**
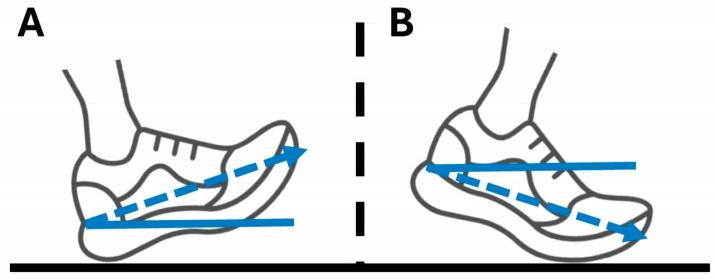
**Typical rearfoot (A) and forefoot (B) foot strike pattern, with example foot strike angles overlaid (blue lines)**.

**Figure 2 sensors-25-02656-f002:**
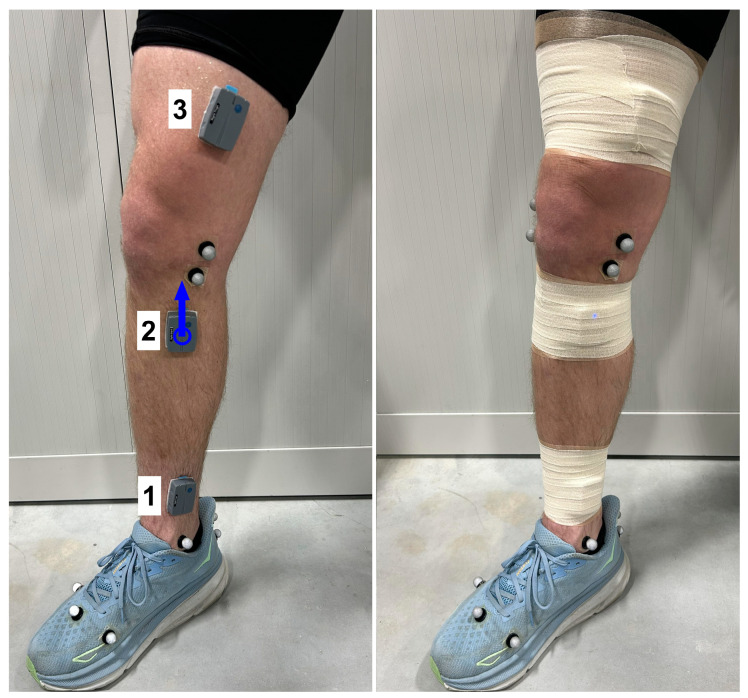
**A depiction of inertial measurement unit placement on lower extremities**. For sensor location categories used in the analyses, 1 = ankle, 2 = below knee, and 3 = above knee. The blue arrow indicates an example vertical axis for IMUs. An adhesive spray was applied to the skin, sensors were adhered with double-sided tape, and then pre-wrap and Coban tape were used to secure them and limit artifact movement.

**Figure 3 sensors-25-02656-f003:**
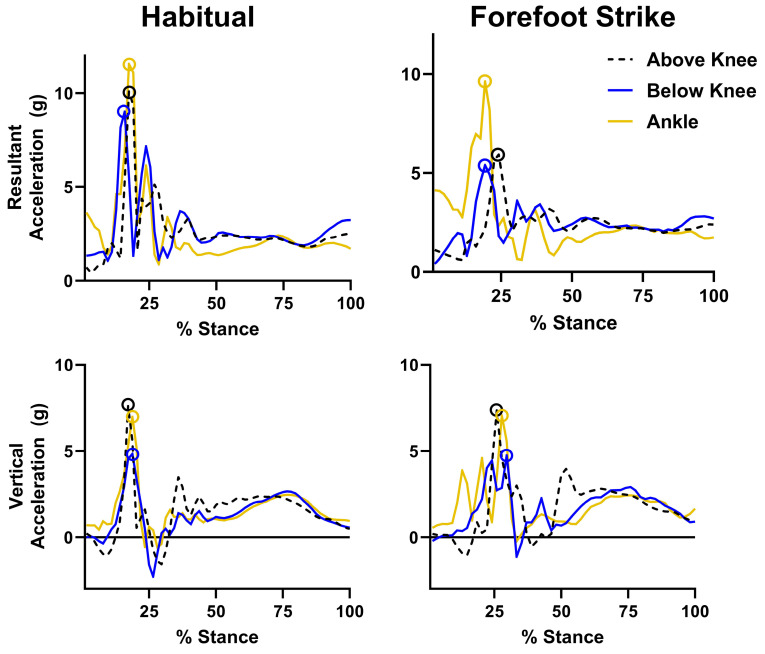
**Example waveforms for accelerations from three sensor locations x gait condition**. Peak accelerations marked with open circles for each waveform.

**Figure 4 sensors-25-02656-f004:**
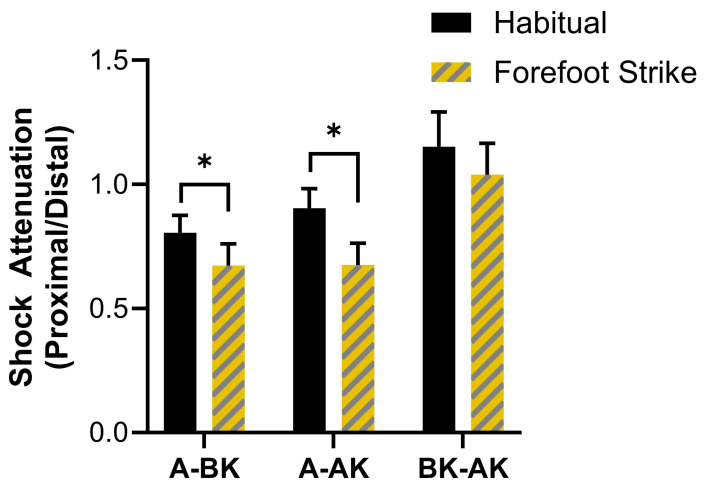
**The results for resultant peak acceleration shock attenuation between sensor locations and gait conditions**. Error bars represent the 95% confidence interval. * indicates significant pairwise comparison (Bonferroni-adjusted *p*-value < 0.05).

**Figure 5 sensors-25-02656-f005:**
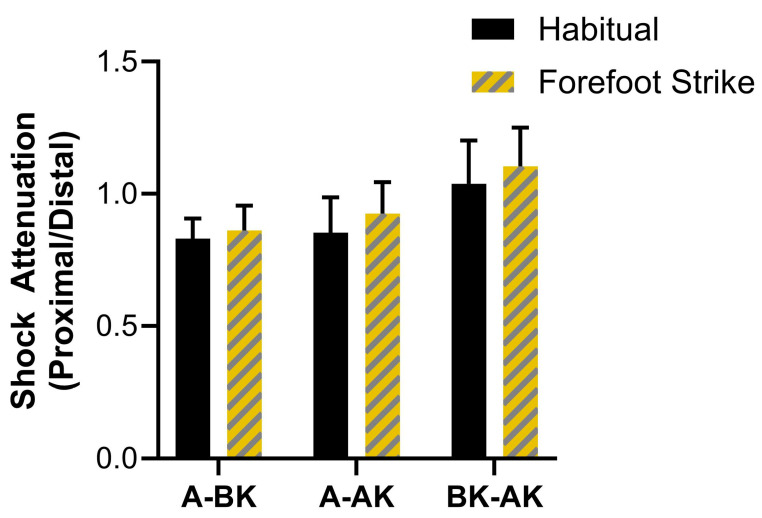
**The results for vertical peak acceleration shock attenuation between sensor locations and gait conditions**. Error bars represent the 95% confidence interval. All pairwise comparisons are non-significant (Bonferroni-adjusted *p*-value ≥ 0.05).

## Data Availability

Sharing of the data underlying this article is restricted by MRDC regulations to institutions/individuals with appropriate data sharing agreements with the USARIEM.

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
