# Peer review of "The Effects of Decreasing Foot Strike Angle on Lower Extremity Shock Attenuation Measured with Wearable Sensors"

_sensors, 2025, doi:10.3390/s25092656_

Round 1

Reviewer 1 Report

Comments and Suggestions for Authors

This study investigated the impact of reducing the foot strike angle (forefoot landing) on shock attenuation in the lower extremities, using wearable sensors to measure changes in shock attenuation under different gait conditions. The research design was reasonable, the methods were rigorous, and the data analysis was comprehensive, demonstrating certain innovation and clinical significance. However, there is still room for improvement in some details and discussions of the article. Comments on the manuscript review are as follows:

  1. In the research background section, it is recommended to add more literature support regarding the relationship between shock attenuation and joint load, especially the specific impact of forefoot landing on the load of the knee and ankle joints, to better introduce the research hypothesis. Clearly point out that previous studies have not involved the measurement of local shock attenuation in the lower limbs, thereby highlighting the innovation of this study.
  2. Although Figure 1 describes the sensor positions, the text part can provide more detailed information on how to ensure the stability of the sensors during movement, especially how to avoid displacement caused by changes in gait.
  3. This study uses acceleration sensors. It is recommended to include the sensor model and working principle in the text and supplement the initial measurement data in the subsequent text to enhance the reliability and accuracy of the conclusion.
  4. The error bars in Figures 2 and 3 represent the 95% confidence interval. It is recommended to clearly mark this in the figure or figure caption. Additionally, it is suggested to modify the colors of Figures 2 and 3 to enhance the contrast of the images.
  5. The text mentions that there is no significant difference in vertical shock attenuation, but the possible reasons are not fully explained. It is recommended to conduct further analysis in combination with sensor axial alignment or gait biomechanics.
  6. To enrich the content of the article, it is recommended to cite the following references: DOI: 10.1016/j.cej.2024.152281.

Reviewer 2 Report

Comments and Suggestions for Authors

The topic is interesting. The manuscript is also well written. However, authors should explain some issues more details. Furthermore, some important information should be added to the manuscript before it can be accepted for publication. My comments are as follows.

  1. Could you add the definition of foot strike angle using a figure?
  2. It would be better if there is a figure that explains the difference between forefoot striking and habitual gait. Do you have any information regarding the percentage of habitual forefoot strikers?
  3. Regarding the IMUs, what kind of sensors included in the devices?
  4. Line 116 and 119: Different units were used.
  5. Line 130 to 134: Which axis is the vertical axis of three axes? Adding a figure should help reader to understand.
  6. Line 140: Is there any literature that support the set threshold of 40N?
  7. In Results, please provide a sample of acceleration waves of three IMUs obtained from the experiment. Could you indicate the differences between habitual gait and forefoot striking, that can be seen from those acceleration waves.
  8. How many trials did you conduct for each volunteer? Did you consider the reproducibility among trials as well as among volunteers?
  9. Three female volunteers participated in the experiment. Was there any effect on the results due to gender difference? Furthermore, bias in the number of volunteers is questionable.
  10. Difference in weight of volunteer caused difference in acceleration magnitude. Did you consider this matter? I thought that some kind of normalization is required during data reduction and analysis.
  11. A-BK, A-AK, BK-AK is confusing. A/BK, AK/A, AK/BK is better notation for proximal/distal.
  12. Shock attenuation ratios for BK-AK were over one. I thought that this is due to the whip motion effect. Considering that foot and calf were constrained during stance phase, thigh was greatly accelerated, causing increase in acceleration response. Please take into consideration.

Round 2

Reviewer 1 Report

Comments and Suggestions for Authors

Thank you for carefully addressing the comments raised in the first round of review. The quality of this manuscript has significantly improved. I support the publication of this manuscript with no further comments on its contents.

Reviewer 2 Report

Comments and Suggestions for Authors

Thank you for your response.

My recommendation is accepted for publication.